# Depressive Symptoms After PCB Exposure: Hypotheses for Underlying Pathomechanisms via the Thyroid and Dopamine System

**DOI:** 10.3390/ijerph16060950

**Published:** 2019-03-16

**Authors:** Petra Maria Gaum, Monika Gube, André Esser, Thomas Schettgen, Natalia Quinete, Jens Bertram, Franziska Maria Putschögl, Thomas Kraus, Jessica Lang

**Affiliations:** 1Institute for Occupational, Social and Environmental Medicine, RWTH Aachen University, Pauwelsstraße 30, 52074 Aachen, North Rhine Westphalia, Germany; monika.gube@staedteregion-aachen.de (M.G.); anesser@ukaachen.de (A.E.); tschettgen@ukaachen.de (T.S.); nsoaresq@fiu.edu (N.Q.); jbertram@ukaachen.de (J.B.); Franziska.Putschoegl@zi-mannheim.de (F.M.P.); tkaus@ukaachen.de (T.K.); jlang@ukaachen.de (J.L.); 2Health Authority of the City and Area of Aachen, Trierer Straße 1, 52070 Aachen, Germany; 3Florida International University, Southeast Environmental Research Center, Modesto A. Maidique Campus, 11200 SW 8th Street, ECS 450, Miami, FL 33199, USA; 4Department of Psychiatry and Psychotherapy, Central Institute of Mental Health, J 5, 68159 Mannheim, Germany

**Keywords:** polychlorinated biphenyls, depression, homovanillic acid, free T4, occupational exposure, pathomechanism

## Abstract

Polychlorinated biphenyls’ (PCB) exposure has been reported to be associated with depressive symptoms, which is correlated to lower dopamine- (DA) and thyroxine-concentrations (T4). T4 is necessary for DA-synthesis and it binds to transthyretin (TTR) being transported into the brain. PCBs can displace T4 by binding to TTR itself, being transported into the brain and disturbing DA-synthesis, where depressive symptoms might occur. Consequently, the free T4-concentration (fT4) increases when PCBs bind to TTR. The interaction of PCBs with fT4 and its associations with the main DA metabolite, homovanillic acid (HVA), and depressive symptoms were investigated. In total, 116 participants (91.6% men) were investigated, who took part in three annual examinations (t1–t3) of the HELPcB health surveillance program. Blood was collected for measuring PCBs, hydroxy PCBs (OH-PCBs), and fT4 and urine for HVA. Depressive Symptoms were assessed with a standardized questionnaire. Interactions were tested cross-sectionally with multiple hierarchical regressions and longitudinally with mixed effect models. Related to HVA, an interaction was cross-sectionally found for lower-chlorinated PCBs (LPCBs) and dioxin-like PCBs (dlPCBs); longitudinally only for LPCBs. Related to depressive symptoms, the interaction was found for LPCBs, dlPCBs, and OH-PCBs; longitudinally again only for LPCBs. The results give first hints that a physiological process involving the thyroid and DA system is responsible for depressive symptoms after PCB exposure.

## 1. Introduction

Many hazardous substances occur in our environment and might pose a dose-dependent risk to human health. Polychlorinated biphenyls (PCBs) are a group of such substances. In the last century, many industrial sectors used PCBs, for example, as a dielectric in transformers and capacitors [1]. Although PCBs were banned [2,3], they are still of great concern due to their high persistence. Thus, PCB exposure of the general population is present in developed, as well as in developing, countries [4]. According to the degree of chlorination, the resulting path of exposure (e.g., via nutrition or inhalation), and their chemical structure (non-coplanar vs. coplanar), PCBs can be classified into three separate groups: lower-chlorinated PCBs (LPCBs), higher-chlorinated PCBs (HPCBs), and dioxin-like PCBs (dlPCBs). LPCBs have five or less chlorine atoms and are typically associated with occupational exposure and exposure via the inhalation of contaminated air in buildings [5]. LPCBs can be metabolized in humans and diminished in the environment and are therefore not detectable with ambient monitoring. HPCBs have more than five chlorine atoms and typically represent environmental exposure via the food chain [6]. The focus of this study according to LPCBs, as well as HPCBs, is the degree of chlorination. In the third group, the chemical structure is focused; the coplanar dlPCBs with a similar chemical structure to dioxins. Related to PCB exposure, many negative health consequences such as skin diseases [7], changes in thyroid function [8], or cancer [9] are reported. Furthermore, previous studies also show negative consequences for mental health (e.g., [10]). The most consistent findings are related to depression and depressive symptoms after occupational (e.g., [10,11]), as well as environmental PCB exposure (e.g., [12]). Prior findings on possible mechanisms to explain depressive symptoms after PCB exposure are rare. In this study, we focus on the toxic effects of PCBs on the nervous system (i.e., the dopamine system), as well as the effects on the thyroid function, to consider a possible mechanism.

The first considered approach for an underlying pathomechanism between PCB exposure and depressive symptoms is related to the central dopamine (DA) system. DA, as well as serotonin and norepinephrine, are neurotransmitters of the monoaminergic system. The monoamines in the central nervous system play an important role in the development of depression. In depressive patients, lower levels of these neurotransmitters were found compared to healthy control groups (e.g., [13]), indicating that there is a negative association between DA levels and depressive symptoms. A great number of animal studies (e.g., [14]), as well as human studies (e.g., [15]) show that the neurotransmitter system of DA is affected by PCBs. PCB intervenes in various ways in the DA system; it may interfere with the synthesis of DA via disturbing tyrosine hydroxylase activity [16], the transport of DA from the synaptic cleft back into the synapsis via blocking the DA transporter [15], and the inhibition of the DA transporter by influencing DA metabolism [17,18]. A prior study of the HELPcB population found a negative association between each type of PCB and the main metabolite of DA, homovanillic acid (HVA), directly after the end of PCB exposure [17]. A further study found that the association between PCB body burden and the number of reported depressive symptoms one year after exposure was mediated by HVA [19]. These results indicate that the influence of PCBs on HVA as a proxy for DA influences the amount of depressive symptoms one year after exposure. In addition to the approach of a DA-related mechanism, we focus on an extended mechanism via thyroid hormones, because PCBs disturb the thyroid system and the thyroid system can interact with the DA system.

With regard to PCBs, various studies reported that PCB exposure alters thyroid function in both directions. Bloom et al. report a negative association of thyroid-like PCBs (28, 52, 60, 74, 77, 95, 99, 101, 105, 114, 118, and 126) with total triiodothyronine (T3), as well as fT4 [20]. Additionally, lower levels of fT3 and fT4 were reported for PCB exposed humans compared to a non-exposed control group [21]. In contrast, positive associations were reported between different PCB congeners and fT4 in fish eaters [22]. In a prior study, we found an association of PCB with lower levels of free T3 over a period of three years after PCB-exposure [8]. In this previous study, we found changes in thyroid function after PCB exposure that might be involved in the development of depressive symptoms.

In non-PCB exposed humans, previous studies reported that serum fT4 can be an indicator for fT4 concentration in the brain [23]. When trying to link thyroid hormone levels and depressive symptoms in patients with hypo- or hyperthyroidism, more depressive symptoms occurred compared to healthy controls [24]. Similarly, Berent et al. [25] reported a positive association of fT4 with the improvement of depression. However, thyroid hormones have been elevated in studies with depressive humans [25,26]. Thus, there are contradictory outcomes, depending on the target group in question (depressive patients vs. patients with thyroid disorder).

In general, however, there seems to be a link between the thyroid system and the DA system, which is also present in depressive disorders; the thyroid system can be affected by DA and the DA system by thyroid hormones. In mice, DA inhibits the release of the thyroid hormone thyroxine [27]. Further animal studies report an elevation in DA level after T4 injection [28,29] and vice versa, and a lower DA level in experimentally-induced hypothyroidism [30]. Hassan et al. [29,30] have demonstrated an important role of T4 in the synthesis of DA. If there is too little T4 in the brain, insufficient DA can be synthesized or released, leading to more depressive symptoms. However, T4 is only active in its free from (fT4). In humans, the majority of T4 is bound to transport proteins (95%–99%), such as TBG (thyroxin-binding globulin, 75%), TTR (transthyretin, 20%), or albumin (5%) [31]. It is assumed that TBG is responsible for T4 transport in the body, while TTR is supposed to pass the blood-brain barrier and transports T4 into the brain [32]. Prior findings confirm this mechanism and show that the TTR concentration in cerebrospinal fluid (CSF) is relatively high compared to other proteins [33]. Patients with major depression have a lower TTR level in the CSF than healthy controls [34]. PCBs have a similar chemical structure to T4 [35] and therefore, some PCBs have a higher affinity to bind with TTR than T4 itself [36]. Additionally, hydroxy-PCBs (OH-PCBs), as the main metabolites of PCBs, have an even higher affinity to bind on TTR than the parent congeners [37]. In the case that PCBs or OH-PCBs bind to TTR rather than T4, one can assume that there is more fT4 in the blood, while less T4 can be transported into the brain and the synthesis of DA is disturbed. The concentration of DA decreases and typical symptoms of depression may occur.

The aim of this study is to investigate the interaction between fT4 levels and PCB exposure as one possible physiological underlying mechanism to explain the occurrence of depressive symptoms. Two main interaction hypotheses will be tested for several types of PCBs.

With regard to non PCB-exposed humans, and in consideration of past literature, we assume that there is a positive association between ft4 and the main DA metabolite HVA. In the case of a high PCB blood concentration, a negative association is postulated. Therefore, an interaction hypothesis with opposite directions of the simple slopes was postulated. The negative association between fT4 and HVA exposure is expected in the case of high PCB blood concentration, which means that a higher PCB and OH-PCB blood concentration should be accompanied with a negative association; so that higher concentrations of fT4 are associated with a lower HVA concentration. In contrast, a positive association between fT4 and HVA is expected in the case of no PCB exposure. Humans with low or no PCB body burden should show a positive association where a low fT4 level is accompanied with a lower HVA concentration. To summarize, there is an interaction between PCBs and fT4 related to HVA. We expect that the correlation between fT4 and HVA is negative in a high PCB blood concentration and positive in a low and no PCB blood concentration (interaction hypothesis 1). We expect the same interaction for OH-PCBs. Since depressive symptoms are associated with a low HVA concentration, this interaction should be inverse to the first postulated interaction. In the case of high PCB body burden, a high fT4 level should be associated with more depressive symptoms and in the case of no or low PCB body burden, a high fT4 level should be associated with fewer depressive symptoms. We suppose in interaction hypothesis 2 that there is a positive correlation between fT4 and depressive symptoms in high PCB exposure and a negative correlation in low or no PCB exposure. We again suspect the same interaction for OH-PCBs, because OH-PCBs are highly correlated with the parent PCB congeners [38]. A graphical illustration of the postulated interaction hypotheses is presented in Figure 1.

## 2. Materials and Methods

### 2.1. Study Design

This is an observational study with a three-wave longitudinal within-subjects-design and a time lag of one year between each examination (t1–t3). All participants were examined as part of the long-term HELPcB (Health Effects in high Level exposure to PCB) surveillance program. This program focuses on PCB-exposed workers of a recycling company, surrounding companies and their relatives. Each year, every participant received an invitation letter to a medical examination and an additional reminder of it by phone. Within a predefined timeframe (09:00 a.m.–11:00 a.m.), blood and spot urine samples were collected, a medical interview was conducted, and every participant was screened for mental syndromes during the medical examination in the outpatient clinic of the study center. A detailed description of the HELPcB program including exposure, participant recruitment, and eligibility criteria is illustrated in [2]. As part of the HELPcB program, this study was approved by the Ethics Commission of the Medical Faculty of the RWTH Aachen University (no. EK 176/11).

### 2.2. Study Population

In total, 300 individuals participated at least once in the HELPcB program and 152 participated on all three measurement occasions. Of these participants, 36 were excluded. Reasons for exclusion were non-participation in the questionnaire due to language problems (*n* = 3), thyroid-relevant medication (*n* = 14), antidepressants (*n* = 12), contraceptives (*n* = 4), pregnancy (*n* = 1), and other relevant medication (i.e., hormone therapy or Parkinson medication; *n* = 2). Altogether, 116 participants were included in the present analyses. Thereof, 106 (91.4%) were males and 10 (8.6%) were females. A statistical description of all used variables can be found in Table 1.

### 2.3. Data Collection

#### 2.3.1. Polychlorinated and Hydroxylated Biphenyls

PCBs and their hydroxylated metabolites OH-PCBs were measured in human plasma samples. Schettgen et al. [5,39] and Quinete et al. [38] described in detail the methodologies for the determination of PCBs and OH-PCBs, respectively. Altogether, 18 PCBs were measured: six indicator PCB congeners and twelve dioxin-like PCBs. Furthermore, 19 hydroxy-PCBs (OH-PCBs) were measured. In four dlPCBs (77, 81, 126, and 169) and six OH-PCBs (4-OH-CB3, 4-OH-CB9, 4-OH-CB15, 4-OH-CB18, 3-OH-CB101, and 4-OH-CB130), less than 20% of the participants showed an interpretable value above the limit of detection/quantitation, so they had to be excluded from all analyses. The remaining 14 PCB congeners were summed up into three variables according to their path of exposure and their chemical structure: lower chlorinated PCBs (LPCBs: PCB 28, PCB 52, PCB 101), higher chlorinated PCBs (HPCBs: PCB 138, PCB 153, PCB 180), and dioxin-like PCBs (dlPCBs: PCB 105, PCB 114, PCB 118, PCB 123, PCB 156, PCB 157, PCB 167, PCB 189). Additionally, 13 OH-PCBs were summed up to an OH-PCB sum variable (3-OH-CB28, 4-OH-CB61, 4-OH-CB76, 4-OHCB101, 4-OH-CB107, 4-OH-CB108, 3-OH-CB118, 3-OH-CB138, 4-OH-CB146, 3-OH-CB153, 4-OH-CB172, 3-OH-CB180, and 4-OH-CB187). For OH-PCBs, the sum of the free phenolic forms and the glucuronide and sulfate conjugates were determined by submitting the samples to enzymatic hydrolysis before extraction. Briefly, 100 µL of plasma went through enzymatic hydrolysis to release the target compound from plasma, followed by protein precipitation, and was analyzed by an online solid phase extraction method coupled to liquid chromatography-tandem mass spectrometry (LC-MS/MS) in multiple reaction monitoring mode (MRM) [42]. A detailed description of the OH-PCB detection can be found in the appendix at the end of this document (Appendix B).

#### 2.3.2. Total Lipids

Since PCBs are lipophilic, blood levels of lipids were measured to ensure a good interpretability of the results. In the serum, cholesterol and triglycerides were detected and the short formula from the CDC (Centers for Disease Control and Prevention) was used to calculate total lipid levels: total lipids = (2.27 × total cholesterol) + triglycerides + 62.3mg/dL [41]. The total lipids were used to adjust the measured lipophilic PCBs for blood lipid level (ng/g lipid).

#### 2.3.3. Free Thyroxine

Free thyroxine (fT4) was analyzed in the serum via an electrochemiluminiscent immunoassay (ECLIA). First, a 15 µL sample was incubated with a ruthenium complex of marked T4-specific antibodies. After adding biotinylated T4 and the reagent streptavidin-coated microparticles to form an antibody-hapten complex, the free binding sites of the marked antibody were occupied. The resulting complex was bound to the solid phase via the biotin-streptavidin interaction. The reaction mixture was then transferred to the measuring cell, where the microparticles were fixed to the surface of the electrode by magnetic action. Afterwards, the unbound substances were removed with ProCell/ProCell M. The microparticles were then removed from the electrode. By applying a voltage, the chemiluminescence emission was induced and measured with the photomultiplier. The results were determined on the basis of a calibration curve. The laboratory internal reference values were 12–22 pmol/L.

#### 2.3.4. Homovanillic Acid

HVA, as the main metabolite of DA, is used as an indicator to map central DA concentration [43,44]. Only 12% of urinary HVA originates from the brain [44], but nevertheless, urinary HVA was used as an indicator for DA because of the non-invasive method and due to the examination design. We collected random urinary samples and urinary HVA concentrations were detected via HPLC high performance liquid chromatography. Jaffe color reaction was used to analyze urinary creatinine concentrations and values are expressed as the ratio of HVA to creatinine (HVA/Crea; in µmol/g creatinine) in order to adjust the measurement for individual urine density [45].

#### 2.3.5. Depressive Symptoms

In order to detect depressive symptoms, we used the German version of the patient health questionnaire (PHQ-D) [46]. The PHQ-D is a multiple-choice self-report inventory and consists of nine items scoring for the DSM-related criteria of depression. “Little interest or pleasure in doing things” or “Feeling tired or having little energy” are two sample items. The participants were asked how often they felt bothered by the considered depressive symptoms in the last two weeks (0 = ”not at all” until 3 = “nearly every day”). A sum scale was built with a possible range from zero to 27. The final scale had a good standardized Cronbach’s alpha in all cross-sections: t1: 0.88, t2: 0.90, t3: 0.91.

### 2.4. Statistical Analyses

First, to identify important confounding variables, a directed acyclic graph was generated [47] with the online tool DAGitty v3.0 [48] (see Appendix A). The minimal sufficient adjustment set consists of age, weight, and albumin. Age is associated with PCB body burden, because of the long half-life of PCBs. Age is also associated with the concentration of HVA and depression [13]. Weight is associated with HVA [44] and changes in thyroid hormone concentration can affect it [49]. Albumin was added as a control variable, because it is a marker for liver function [50]. Liver function has an effect on the concentration of fT4 and HVA as these are metabolized in the liver. Finally, it was controlled for gender, because women have a higher prevalence of depression [51] and a lower concentration of PCB body burden, since they are not directly exposed in their job [52]. Gender was also controlled for the analyses with OH-PCBs, because the OH-PCB body burden is caused by the metabolism of PCBs and LPCBs, which are typical for job exposure and have a shorter half-life than HPCBs.

Both interaction hypotheses were tested in two steps. First, the postulated interactions for the individual cross-sections were considered separately. Then, a mixed effect model was calculated in which the cross-section was included as a random effect. Following this procedure made it possible to a) control and investigate the influence of time, b) to test whether the interactions are stable over the three years, and c) to account for the problem of multiple testing. SPSS 25 for Windows [53] and SPSS macro PROCESS [54] were used to calculate the separate interactions for the respective cross-sections. For the analysis of the mixed effect models, the statistic program R version 3.4.3 [55] and RStudio version 1.1.383 [56] with the package “lmerTest” [57] were used. The lipid adjusted PCB and sum variables and the OH-PCB sum variable were naturally log transformed to reach a normal distribution of the residuals for regression analyses. Further, all variables were z-standardized to get standardized beta-coefficients in the outputs. To visualize the interactions of the mixed effect models, the R package “lattice” was used [58]. According to Baron and Kenny [59], no direct effect between the predictor and the mediator is necessary to test an interaction hypothesis. Nevertheless, we provide the direct associations between several PCB congeners, as well as OH-PCBs, and fT4 in Table 2. All hypotheses were tested one-sided, since they were directed hypotheses. The level of significance for one-sided testing was set to a *p*-value lower than 0.05. 

## 3. Results

The first interaction hypothesis was related to PCBs’ and OH-PCBs’ impact on the association between fT4 and the dopamine metabolite HVA. In the cross-sectional analyses, we find significant interaction terms for LPCBs at t2 and t3 and for dlPCBs at t2 (see Table 3). No interaction terms are significant for HPCBs and OH-PCBs at each examination. The three significant cross-sectional interactions are illustrated in Figure 2. Cross-sectional results only partially confirm the postulated interaction hypothesis.

The longitudinal results with mixed models only partially support the prior tested cross-sectional hypotheses. According to the first postulated interaction related to HVA/crea, a significant interaction was only found for LPCBs (see Table 4). The interaction for LPCBs is illustrated in Figure 3a. No significant interactions are found for HPCBs (β = −0.07, *t* = −1.09, *p* = 0.14) and dlPCBs (β = −0.09, *t* = −1.40, *p* = 0.08), as well as for OH-PCBs (β = −0.06, *t* = −0.92, *p* = 0.18). There is only a significant interaction of LPCBs on the association between fT4 and HVA/crea, so hypothesis one can only be partially confirmed.

In the second interaction hypothesis, the moderating effect of PCBs and OH-PCBs on the association between fT4 and the amount of depressive symptoms was tested. The cross-sectional analyses show significant interaction terms for LPCBs at t1 and t2, as well as for dlPCBs at t2 and OH-PCBs at t1 (see also Table 3). These four significant interactions are illustrated in Figure 4. No significant interaction term was found for HPCBs at all examinations. Thus, the cross-sectional interaction hypothesis related to depressive symptoms is only partially confirmed.

In the mixed effect model analyses, only the interaction effect of LPCBs is significant (see Table 4). The significant interaction effect for LPCBs is visualized in Figure 3b. For HPCBs (β = 0.02, t = 0.39, *p* = 0.35), dlPCBs (β = 0.07, *t* = −1.24, *p* = 0.11), and OH-PCBs (β = 0.07, *t* = 1.42, *p* = 0.08), no significant interactions with fT4 on depressive symptoms are found. Thus, hypothesis two can also be partially confirmed for LPCBs under control for the measurement occasions.

## 4. Discussion

In this study, we investigated one possible physiological pathomechanism to explain the positive association between PCB exposure and depressive symptoms [10]. An approach via the thyroid hormone transporter TTR and the impact on the DA system and on depressive symptoms derived from the literature and cross-sectionally, as well as longitudinally, was tested with two hypotheses.

In our first hypothesis, we postulated an interaction of PCB with fT4 on the dopamine metabolite HVA. The association between fT4 and the dopamine metabolite HVA is normally positive, but when PCBs bind to TTR, the association inverts and a high fT4 concentration is associated with lower HVA. Cross-sectional results show the postulated interaction for LPCBs and dlPCBs at t2, but at t3, the interaction was in the opposite direction to the postulated interaction for LPCB. One reason for this could be the decrease in PCB concentrations over time. Since the participants in this study sample are highly contaminated with PCBs, the concentration of almost all types of PCBs decreases over time (see Table 1). This could reduce the influence of PCBs on the association between fT4 and HVA. However, it should be noted that the results can only be correctly interpreted if the random influence of the measurement occasion has been controlled. In the mixed effect analyses, the interactions were found in the postulated direction, specifically for LPCBs.

The second hypothesis focused on depressive symptoms. Since patients with diagnosed depression have a lower HVA concentration than healthy controls [60], the second hypothesis expected opposite associations; a negative association between fT4 and depressive symptoms without PCB exposure and a positive one with PCB exposure. The cross-sectional results show the postulated interaction for LPCBS and OH-PCBs at t1 and for LPCBs and dlPCBs at t2. According to longitudinal effects, the postulated interaction was found again only for LPCBs after controlling for random effects of the measurement occasions.

In vitro studies show that dlPCBs [35] and OH-PCBs [61] have the highest affinity to TTR. This would mean that dlPCBs exhibit stronger binding to TTR than the non-dlPCBs and OH-PCBs exhibit stronger binding than the parent congeners. Interestingly, although LPCBs have the lowest affinity, our results show only significant effects for LPCBs on HVA and depressive symptoms. One reason why significant interactions for HVA and depressive symptoms were found only for LPCBs could be due to the type of TTR. The literature describes that there is a difference between TTR in the central nervous system and peripheral TTR in the body [34]. There are findings which show that CSF-TTR is synthesized in the central nervous system (i.e., choroid plexus), independently of TTR in the blood [32]. Therefore, a disorder in the peripheral thyroid system does not necessarily have to be related to a disorder in the CSF-TTR-T4 system and vice versa [35]. Lans et al. [61] and Chauhan et al. [33], however, did not use the central TTR, but the TTR from the blood, for affinity analysis of PCBs. The different types of PCBs could have different binding properties towards CSF-TTR and this could be a possible explanation for why the hypotheses in this study could only be confirmed for LPCBs. A further explanation may be the estrogen-like activity of LPCBs. LPCBs show the strongest estrogen-like activity and the highest affinity to bind to estrogen receptors compared to other PCBs [62]. OH-PCBs only have a very low affinity to bind to estrogen receptors [63]. Estrogen can have effects on both the concentration of thyroxine transporters [64] and the dopamine system [65]. Additionally, the postulated mechanism is only one of several possible mechanisms to explain depressive symptoms after PCB exposure. An interaction hypothesis was tested in this study, but according to the literature, a causal association via a mediator would also be a possible mechanism, because PCBs can alter the DA system [29,30] and DA affects the thyroid system [27]. Another possible reason why the interaction was only significant for LPCBs may be that the group of LPCBs is the only PCB group in this study that only consists of thyroid-like PCBs [20]. Many studies report associations between PCB exposure and thyroid-related outcomes; a negative association between PCB 101 and 149 with fT4 [66] or a positive association of PCBs with fT4 [67]. Since the focus of this study was not solely on the thyroid gland, thyroid-like PCBs were not explicitly determined. Because of this, no PCB with all thyroid-like congeners could be created. Nevertheless, a further possible reason for the weak effects could also be that people exposed to PCBs are never exposed to just one PCB. There is always mixed exposure.

The interactions found are weaker in relation to HVA than in relation to depressive symptoms regarding effect sizes of the longitudinal analyses. Sullivan et al. [35] describe that TTR as a tetramer polypeptide cannot pass the blood-brain barrier, but PCBs can pass the blood-brain barrier without a transporter because they are lipophilic [68]. PCBs can also change the permeability of the blood-brain barrier [69]. When PCBs now pass the blood-brain barrier (BBB) and dock to the CSF-TTR, this may have the same effect as previously described for plasma TTR. CSF-TTR generally transports T4 via the CSF to the brain areas (e.g., cortex or striatum), where it is needed for the synthesis of neurotransmitters (i.e., DA) [32]. In case PCBs dock to TTR, the transport of T4 to the neurons is disturbed and also the synthesis of DA. The fact that PCBs can pass the blood-brain barrier without a transporter supports the theory that the findings of this study could be a central process. Since only 12% of urinary HVA originates from the brain [44], this could explain why the effects for HVA were weaker than for depressive symptoms. If we assume a central process, then only central HVA can reflect the postulated mechanism. However, the measured HVA values consist of central HVA and peripheral HVA, which can lead to a lower effect power, supporting the results found in our study. It is important to note that this study mainly consisted of male workers, which could be considered one of the study weaknesses. A direct negative correlation between CSF-TTR and depression was only found for men; in women, the correlation was not significant [34]. This could also explain the stronger effects related to depressive symptoms. Another reason may be the measurement of depressive symptoms with the PHQ; a questionnaire with high validity and very good reliability [70].

As described before, since only 12% of peripheral HVA derives from the central nervous system, the study is limited to assessing the central neurotransmitter system in total [43,44]. Furthermore, for HVA measurement, certain confounding factors, such as the consumption of foods with a high monoamine content and individual variability, have to be considered. Given these limitations, the peripheral measurement of HVA via urinary HVA detection can still be used as a direct, practical, and non-invasive method to map changing concentrations of central DA [44].

Another limitation is that we did not measure TTR directly. Instead of TTR, the concentration of fT4 in the blood was indirectly measured. In this way, it is expected that if PCBs dock on TTR instead of T4, the blood concentration of fT4 increases proportionally to the concentration of PCBs docking on TTR. Within the HELPcB program, it was not possible to measure the CSF-TTR, because the invasive procedure of a lumbar puncture would have been necessary. Further, the affinity of PCB on TTR was not measured in the study population, so it is not clear which PCB congeners have the highest binding affinity to TTR in our study population. Future studies should investigate thyroid hormone transporters and in particular CSF-TTR in order to explore the mechanism between PCB exposure and depressiveness. Furthermore, it is not clear whether the PCB-related changes in fT4 are strong enough in our study population to have an impact on the DA system. The influence of thyroid hormones on the DA system has only been investigated in animal studies so far [29,30]. However, participants with a thyroid disease were excluded from the analyses to avoid bias of the results by outliers. Besides fT4, other thyroid-related parameters might be relevant to get a clearer picture of the proposed pathomechanism [24]. It might be necessary to also investigate T3 or a thyroid stimulating hormone, to describe a complete mechanism explaining depressive symptoms after PCB exposure. Future studies should take these additional hormones into account in their analyses.

## 5. Conclusions

The described results in this study are not clear enough to support a mechanism via blocked thyroid transporter TTR to explain depression after PCB exposure. However, the results give first hints for a physiological mechanism via disturbed thyroid and DA system to explain the positive correlation between PCB body burden and depressive symptoms. We found an interaction effect predominantly between LPCB exposure and fT4 related to the main DA metabolite HVA, as well as related to depressive symptoms. Human exposure to LPCBs occurs predominantly in an occupational setting and via inhalation in contaminated buildings. This study gives first hints for an underlying pathomechanism that may explain the association of occupational and inhalative PCB exposure and depressive symptoms. The findings show that a physiological process involving the DA and thyroid system may be responsible for depressive symptoms after PCB exposure. Nevertheless, more research is necessary to support these findings. 

## Figures and Tables

**Figure 1 ijerph-16-00950-f001:**
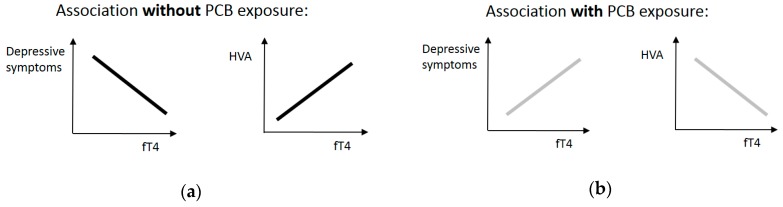
Schematic illustration of the postulated interaction hypotheses. Note: PCB = polychlorinated biphenyls, HVA = homovanillic acid, fT4 = free thyroxin.

**Figure 2 ijerph-16-00950-f002:**
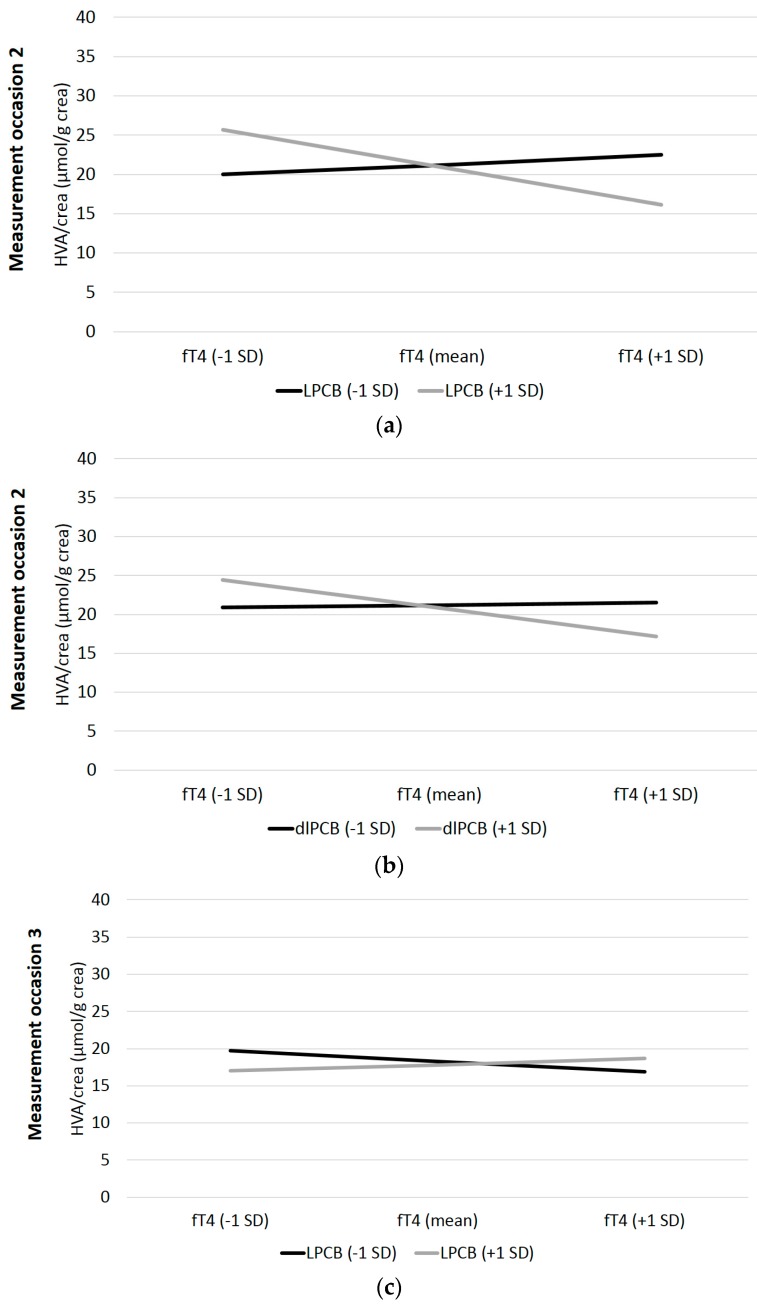
Visualization of the interaction of log transformed lipid adjusted LPCBs (**a** and **c**) and dlPCBs (**b**) on the association of fT4 and HVA; Notes: fT4 = free thyroxine, HVA/crea = homovanillic acid per gram creatinine, SD = standard deviation, LPCB = lower chlorinated PCB, dlPCB = Dixon-like PCB.

**Figure 3 ijerph-16-00950-f003:**
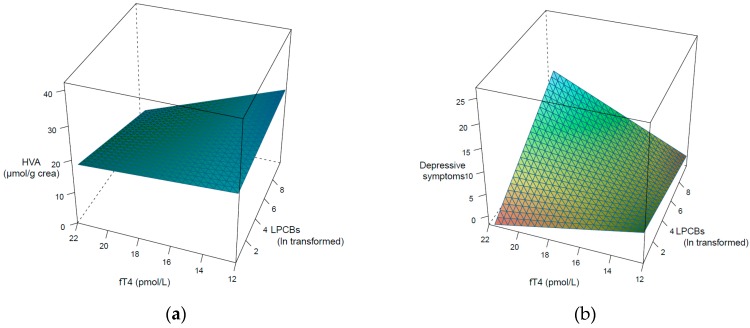
Visualization of the interaction of log transformed and lipid adjusted LPCBs in terms of the association of fT4 with HVA (**a**) and on fT4 with depressive symptoms (**b**).

**Figure 4 ijerph-16-00950-f004:**
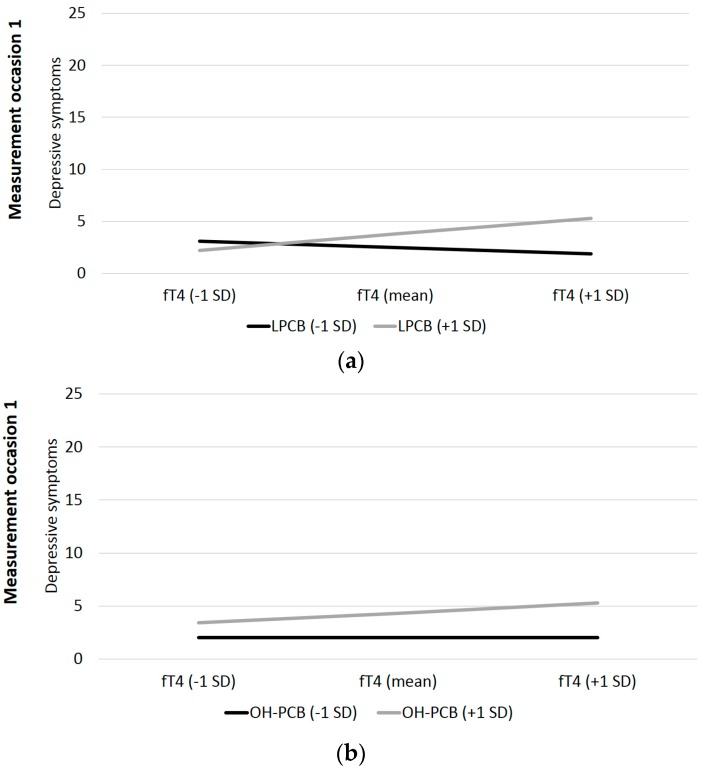
Visualizing the interactions of log transformed and lipid adjusted LPCBs, dlPCBs, and OH-PCBs on the association of fT4 and depressive symptoms; Notes: fT4 = free thyroxine, SD = standard deviation, LPCB = lower chlorinated PCB, dlPCB = dioxin-like PCB, OH-PCB = hydroxylated PCB.

**Table 1 ijerph-16-00950-t001:** Description of the study population (N = 116).

		t1			t2			t3		
	Ref	M (SD)	Mdn	Range	M (SD)	Mdn	Range	M (SD)	Mdn	Range
∑LPCBs (µg/L plasma)	0.02 ^1^	4.6 (21.0)	0.2	<LOD–190.0	3.5 (16.3)	0.1	<LOD–126.4	2.6 (13.1)	0.1	<LOD–114.8
∑HPCBs (µg/L plasma)	2.36 ²	8.9 (19.2)	2.6	0.2–169.1	8.0 (16.8)	2.6	0.3–152.1	8.2 (15.7)	3.0	0.3–135.5
∑dlPCBs (µg/L plasma)	0.05 ^1^	3.4 (8.4)	0.5	0.1–62.4	2.9 (7.1)	0.5	0.1–49.6	2.4 (5.7)	0.4	0.1–36.7
∑OH-PCBs (µg/L plasma)	- ³	5.2 (10.3)	1.7	0.2–71.4	4.6 (10.4)	1.5	0.1–88.8	4.2 (6.0)	1.5	0.1–35.8
fT4 (pmol/L)	12–22 ^4^	15.8 (2.0)	15.7	10.7–20.8	16.0 (2.0)	16.0	11.9–21.6	15.7 (2.0)	15.6	12.2–21.6
HVA/crea (µmol/g crea)	<42 ^4^	19.4 (9.9)	17.9	6.3–79.0	21.0 (8.0)	19.1	11.8–55.6	17.8 (5.7)	17.5	7.1–37.0
Depressive symptoms	0–27	3.2 (3.6)	2.0	0–20	3.5 (3.6)	2.0	0–19	3.7 (4.4)	2.0	0–22
Age (years)	-	44.3 (12.8)	45.0	22–83	45.3 (12.8)	46.0	23–84	46.3 (12.8)	47.0	24–85
Total lipids (g/L serum)	8.6 ^5^	7.6 (2.2)	7.4	4.4–18.9	7.6 (1.9)	7.4	4.2–13.2	7.7 (2.1)	7.3	4.7–15.2
Albumin (g/L serum)	35–52 ^4^	48.1 (2.7)	47.8	41.4–55.4	49.1 (2.7)	48.8	43.4–57.2	48.0 (2.9)	47.7	41.6–54.7
Weight (kg)	-	88.6 (16.5)	85.0	54–140	88.6 (16.0)	85.5	57–145	87.5 (15.5)	85.0	58–125

Notes: N = number of included participants, ref = reference value, t = time (measurement occasion), M = mean, SD = standard deviation, Md = median, PCBs = polychlorinated biphenyls, ∑LPCBs = sum of lower-chlorinated PCBs, ∑HPCBs = sum of higher-chlorinated PCBs, ∑dlPCBs = sum of dioxin-like PCBs, ∑OH-PCBs = sum of hydroxylated PCBs, HVA/crea = homovanillic acid / creatinine, LOD = limit of detection. ^1^ generated with the reported 95th percentiles [39]. ² generated with the reported 95th percentiles [40]. ³ to the best of our knowledge, there are no reference values for OH-PCBs. ^4^ Laboratory internal reference value. ^5^ according to Bernert [41].

**Table 2 ijerph-16-00950-t002:** Significant Spearman rank correlations between PCB congeners and fT4.

PCB	Spearmans Rho	*p*
**t1**		
**lip156**	0.170	0.034
**t3**		
**lip123**	0.187	0.023
**OH-PCB**		
**t1**		
**3OHCB153**	0.208	0.013
**4OHCB172_180**	0.198	0.017
**t3**		
**4OHCB9**	0.201	0.015

Note: PCB = polychlorinated Biphenyls, OH-PCB = hydroxylated Biphenyls, t1/t3 = measurement occasion 1 and 3; *p* = *p*-value (significance, one-tailed), lip = lipidadjusted PCBs (in ng/g lipid). The associations of all other PCB congeners and OH-PCBs with fT4 were not significant.

**Table 3 ijerph-16-00950-t003:** Beta-coefficients for the moderation of lipid adjusted PCBs in terms of the association of fT4 with dopamine metabolite HVA/crea, as well as depressive symptoms at each measurement occasion.

	HVA/Crea					Depressive Symptoms				
Moderation	β	S.E.	*t*	*p*	Δ*R*²	β	S.E.	*t*	*p*	Δ*R*²
t1: LPCB*fT4	−0.03	0.11	−0.30	0.38	0.001	**0.32**	**0.10**	**3.38**	**0.001**	**0.091**
t2: LPCB*fT4	**−0.34**	**0.09**	**−3.77**	**<0.001**	**0.101**	**0.31**	**0.13**	**2.44**	**0.01**	**0.053**
t3: LPCB*fT4	**0.35**	**0.18**	**1.98**	**0.03**	**0.045**	−0.12	0.13	−0.89	0.19	0.009
t1: HPCB*fT4	−0.11	0.12	−0.86	0.19	0.007	0.07	0.10	0.69	0.25	0.004
t2: HPCB*fT4	−0.16	0.09	−1.68	0.05	0.022	0.13	0.12	1.02	0.16	0.009
t3: HPCB*fT4	0.07	0.13	0.49	0.32	0.003	−0.09	0.10	−0.89	0.19	0.008
t1: dlPCB*fT4	−0.08	0.12	−0.64	0.26	0.004	0.13	0.09	1.37	0.09	0.016
t2: dlPCB*fT4	**−0.23**	**0.09**	**−2.51**	**0.01**	**0.048**	**0.20**	**0.12**	**1.69**	**<0.05**	**0.025**
t3: dlPCB*fT4	0.13	0.15	0.84	0.20	0.008	−0.14	0.11	−1.23	0.11	0.015
t1: OH-PCB*fT4	−0.05	0.12	0.41	0.35	0.002	**0.16**	**0.09**	**1.73**	**0.04**	**0.025**
t2: OH-PCB*fT4	−0.09	0.07	−1.22	0.11	0.012	0.15	0.10	1.55	0.06	0.022
t3: OH-PCB*fT4	0.15	0.15	1.03	0.15	0.013	−0.17	0.11	−1.51	0.07	0.023

Notes: controlled for age, weight, albumin, and gender; bootstrapping with N = 5000; PCBs = polychlorinated biphenyls, LPCBs = lower-chlorinated PCBs, HPCBs = higher-chlorinated PCBs, dlPCBs = dioxin-like PCBs, fT4 = thyroxin, t1 = measurement occasion 1, t2 = measurement occasion 2, t3 = measurement occasion 3, β = standardized regression coefficient, S.E. = standard error, *t* = *t*-value, *p* = *p*-value (significance), Δ*R*² = delta *R*square (explained variance by the interaction). Significant interaction terms are in bold.

**Table 4 ijerph-16-00950-t004:** Fixed and random effects of the moderation of log transformed and lipid adjusted LPCBs on the association of fT4 with homovanillic acid and depressive symptoms.

	Homovanillic Acid			Depressive Symptoms		
	β	*t*	*p*	β	*t*	*p*
**Fixed Effect**						
fT4	−0.13	−2.00	0.03	0.04	0.65	0.26
LPCBs	−0.08	−1.32	0.10	0.18	3.35	<0.001
Age	0.11	1.68	0.05	−0.06	−1.14	0.13
Weight	−0.27	−4.55	<0.001	−0.02	−0.23	0.39
Albumin	−0.04	−0.59	0.28	−0.07	−1.23	0.11
Gender	0.47	2.06	0.02	0.44	2.23	0.02
fT4*PCBs	**−0.11**	**−1.72**	**0.04**	**0.19**	**3.37**	**<0.001**
*R*²	0.140			0.104		
**Random Effect** (for measurement occasion)						
Variance	0.05			0.00		
Standard deviation	0.23			0.00		
*R*²	0.050			0.000		

Notes: controlled for age, weight, albumin, gender, and measurement occasion; fT4 = free thyroxine, PCBs = polychlorinated biphenyls, LPCBs = lower-chlorinated Biphenyls, β = standardized regression coefficient, *t* = *t*-value, *p* = *p*-value (significance), *R*² = *R*-squared (explained variance). Significant interaction terms are in bold.

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
