# Peer review of "Depressive Symptoms After PCB Exposure: Hypotheses for Underlying Pathomechanisms via the Thyroid and Dopamine System"

_ijerph, 2019, doi:10.3390/ijerph16060950_

Round 1

Reviewer 1 Report

This paper explores the interesting hypothesis that PCBs increase depressive-scores by binding TTR, thereby reducing thyroid hormone action in the brain, thereby reducing dopamine synthesis. This ties together several lines of PCB mechanisms and should certainly be explored more. The authors correctly identify weaknesses of their own work in that 1) few women are represented, 2) fT4 is an imperfect indicator for thyroid hormone action in the brain, and 3) HVA is an indicator for both peripheral and central dopamine action. These are somewhat unavoidable, given the constraints of sample collection in human studies. However, there are several other major weaknesses that leave me unclear as to the significance of the study as written. The raw data seems important to consider, but interpretation of the data is problematic.

Major concerns:

The authors fail to report main effects of PCB levels on fT4 and HVA; they only state that there are no interactions. But, their hypothesis rests upon the prediction that higher PCB should be associated with higher fT4 because of competitive binding with TTR. If this is not the case, then their explanation for their results seems implausible.

They present the data as a causal relationship, where thyroid hormone drives dopamine synthesis. However, PCBs can alter DA function in many ways independent of thyroid hormone action, AND DA can affect thyroid related endpoints. The authors need to take care to couch their introduction and discussion as one of several possible explanations of a correlation. Moreover, it is unknown if the changes in fT4 are dramatic enough to even alter dopamine synthesis. The only evidence of this phenomenon are in rats, so it is difficult to extrapolate.

If higher levels of thyroid hormone action in the brain (reflected by lower fT4) does drive increased dopamine release (reflected by higher HVA) in a healthy brain, why would the direction of relationship be completely reversed by PCB exposure. Yes, PCBs could bind TTR to increase fT4, but that would just shift the scale of the relationship, not the direction. Same concern for depression outcomes.

Why they did not group their PCBs according to the Hamers 2011 study they cite, as only a few PCBs inhibit TTR?

Missing discussion of the many papers that describe interactions between PCBs and thyroid related endpoints, besides Bloom et al. Also need to discuss other papers that test TTR binding affinities and hepatic clearance of T4.

Other concerns:

Line 51 – it is possible for lower-chlorinated PCBs to be coplanar, but non dioxin like. To be dioxin-like, they must be coplanar AND heavily chlorinated.

Line 74 – the levels quantified in this cited study are not defined clinically as hypo or hyper thyroid.

Line 77 – cite hypothyroid and depression study

Figure 3 – axis of fT4 should be reversed so that low is on left.

Author Response

Dear Reviewer 1,

we thank you very much for your helpful comments regarding our submission. We have revised the manuscript based on the valuable suggestions and advice you made. An item-by-item response to your comments is enclosed. We hope that these revisions successfully address your concerns and requirements.

We thank all of you for your helpful suggestions and your time.

Sincerely,

Dr. Petra Gaum

Reviewer 1 – Item-by-Item-Response

Open Review

English language and style

( ) Extensive editing of English language and style required
(x) Moderate English changes required
( ) English language and style are fine/minor spell check required
( ) I don't feel qualified to judge about the English language and style

Yes

Can be   improved

Must be   improved

Not   applicable

Does the introduction provide sufficient background and include all   relevant references?

( )

( )

(x)

( )

Is the research design appropriate?

( )

(x)

( )

( )

Are the methods adequately described?

( )

(x)

( )

( )

Are the results clearly presented?

( )

(x)

( )

( )

Are the conclusions supported by the results?

( )

( )

(x)

( )

Comments and Suggestions for Authors

This paper explores the interesting hypothesis that PCBs increase depressive-scores by binding TTR, thereby reducing thyroid hormone action in the brain, thereby reducing dopamine synthesis. This ties together several lines of PCB mechanisms and should certainly be explored more. The authors correctly identify weaknesses of their own work in that 1) few women are represented, 2) fT4 is an imperfect indicator for thyroid hormone action in the brain, and 3) HVA is an indicator for both peripheral and central dopamine action. These are somewhat unavoidable, given the constraints of sample collection in human studies. However, there are several other major weaknesses that leave me unclear as to the significance of the study as written. The raw data seems important to consider, but interpretation of the data is problematic.

Major concerns:

The authors fail to report main effects of PCB levels on fT4 and HVA; they only state that there are no interactions. But, their hypothesis rests upon the prediction that higher PCB should be associated with higher fT4 because of competitive binding with TTR. If this is not the case, then their explanation for their results seems implausible.

Thank you for this comment. The main effects of the PCB congeners on HVA in this study population was reported in a prior study of our research group. Please see Putschögl et al. (2015). We now describe the results of this study in the introduction. See lines 71-73: „A prior study of the HELPcB cohort found a negative association between each type of PCBs and the main metabolite of DA, homovanillic acid (HVA) directly after exposure [17].”

There are no significant correlations between the PCB sum variables and fT4 in our study population (Gaum et al., 2016). But there are isolated positive associations between the individual congeners and OH-PCBs with fT4 (see table R1). For the other PCB congeners all correlations were not significant.

Table   R1: Significant Spearman rank correlations between PCB congeners and fT4.

PCB

Spearmans Rho

p

t1

lip156

.170

.034

t3

lip123

.187

.023

OH-PCB

t1

3OHCB153

.208

.013

4OHCB172_180

.198

.017

t3

4OHCB9

.201

.015

Note:   PCB = polychlorinated Biphenyls, OH-PCB = hydroxylated Biphenyls, t1-t3 =   measurement occassion 1 and 3; p = p-value (significance, one-tailed), lip =   lipidadjusted PCBs (in ng/g lipid). 

Nevertheless, an interaction effect can be analyzed although the main effect is not significant.

“The moderator hypothesis is supported if the interaction […] is significant. There may also be significant main effects for the predictor and the moderator […], but these are not directly relevant conceptually to testing the moderator hypothesis. In addition to these basic considerations, it is desirable that the moderator variable be uncorrelated with both the predictor and the criterion (the dependent variable) to provide a clearly interpretable interaction term.” (Baron & Kenny, 1986; p. 1174)

In the case of multicolarity, which would mean that the predictor (fT4) correlates with the moderator (PCB), it would be difficult to interpret the interaction.

They present the data as a causal relationship, where thyroid hormone drives dopamine synthesis. However, PCBs can alter DA function in many ways independent of thyroid hormone action, AND DA can affect thyroid related endpoints. The authors need to take care to couch their introduction and discussion as one of several possible explanations of a correlation.

Please note, we consider the interaction between PCBs and fT4 in this study and no causal association between these variables, such in case of a mediation hypothesis. However, we now discuss the interaction as one possible mechanism in the introduction and in the discussion section. See lines 112-114: “The aim of this study is to investigate this interaction as one possible physiological underlying mechanism to explain the association between PCB exposure and depressive symptoms.” And see lines 345-349: “[…] the postulated mechanism is only one of several possible mechanisms to explain depressive symptoms after PCB exposure. An interaction hypothesis was tested in this study, but according to the literature a causal association via mediator would also be a possible mechanism, because PCBs can alter DA system [23,24] and DA affect thyroid system [25].”

Further, we described the possible impacts of PCBs to the DA system more precisely. See lines 68-71: “PCB intervenes in various ways in the DA system; they may interfere the synthesis of DA via disturbing tyrosine hydroxylase activity [16], the transport of DA from the synaptic cleft back into the synapsis via blocking DA transporter [15], and finally the inhibition of DA transporter also influences DA metabolism  [17,18].”

We also included the information that DA may influence the thyroid system in the introduction. See lines 91f: “In mice DA inhibits the release of the thyroid hormone thyroxine [25].”

Moreover, it is unknown if the changes in fT4 are dramatic enough to even alter dopamine synthesis. The only evidence of this phenomenon are in rats, so it is difficult to extrapolate.

Yes, it is possible that the change in fT4 in our cohort is not strong enough to result in a change in HVA and therefore in depressive symptoms. This is even more probably, because participants with thyroid disease and thyroid-related drugs were excluded. These participants were excluded in order to keep a statistical bias as low as possible. These points have now been discussed in the discussion section. See lines 390 - 394: “Furthermore, it is not clear whether the PCB-related changes in fT4 are strong enough in our study population to have an impact on the DA system. The influence of thyroid hormones on the DA system has only been investigated in animal studies so far [27,28]. However, participants with a thyroid disease were excluded from the analyses to avoid bias of the results by outliers.”

If higher levels of thyroid hormone action in the brain (reflected by lower fT4) does drive increased dopamine release (reflected by higher HVA) in a healthy brain, why would the direction of relationship be completely reversed by PCB exposure. Yes, PCBs could bind TTR to increase fT4, but that would just shift the scale of the relationship, not the direction. Same concern for depression outcomes.

Thank you for this comment. You are right if we would consider total T4 in our study but we used free T4. For fT4 the association is inverted as it increases when PCBs bind to TTR, but the total T4 remains unchanged. We may not have described the mechanism clearly enough. We have now rephrased the relevant information and hope that the mechanism is easier to understand. See lines 117-123 in the introduction: “Therefore an interaction hypothesis with opposite direction of the simple slopes were postulated. A negative association between fT4 and HVA exposure is expected in case of PCB exposure. That means, a higher PCB and OH-PCB blood concentration should be accompanied with a negative association so that higher concentrations of fT4 are associated with lower HVA concentration. In contrast, a positive association between fT4 and HVA is expected in case of no PCB exposure. Humans with low or no PCB body burden should show a positive association that a low fT4 level is accompanied with lower HVA concentration.”

Why they did not group their PCBs according to the Hamers 2011 study they cite, as only a few PCBs inhibit TTR?

Unfortunately the study was originally not focused on binding on TTR. Hamers hat viele verschiedene Kongenere berichtet, die unterschiedliche Affinitäten aufweisen. If these are considered together, then there is a risk that possible effects may not be found. Because not all PCB congeners that were reported in Hamers et al. (2011) were analyzed in the HELPcB program, we decided to create sum variables according to the chemical properties of the considered PCB congeners. Furthermore, the congeners with the strongest affinity were not measured within the HELPcB program.

Missing discussion of the many papers that describe interactions between PCBs and thyroid related endpoints, besides Bloom et al. Also need to discuss other papers that test TTR binding affinities and hepatic clearance of T4.

We included more papers that report associations between PCB and thyroid outcomes.  See lines 82-84 in the introduction: “Also lower levels of fT3 and fT4 in PCB were reported for exposed humans compared to a non-exposed control group [21]. In contrast positive associations were reported between different PCB congeners and fT4 in fish eaters [22].”

And see lines 351-353 in the discussion: “Many studies report associations between PCB exposure and thyroid related outcomes; a negative association between PCB 101 and 149 with fT4 [64] or a positive association of PCBs with fT4 [65].”

Other concerns:

Line 51 – it is possible for lower-chlorinated PCBs to be coplanar, but non dioxin like. To be dioxin-like, they must be coplanar AND heavily chlorinated.

We define lower-chlorinated PCBs with five or less chlorine atoms and higher-chlorinated PCBs with more than five chlorine atoms. Based on this definition, it is possible that some dlPCB have five or less chlorine atoms and thus they are lower-chlorinated (e.g. PCB 77, 81). The place of the substituted chlorine atoms is more relevant for the coplanar structure; especially the ortho positions 2,2’,6,6’. But you are right, that some-lower chlorinated PCBs can be coplanar (e.g. PCb 28).

We now reformulated our categorization of the several types of PCBs more precisely. See lines 47-53: “LPCBs have five or less chlorine atoms and are typically associated to occupational exposure and exposure via inhalation of contaminated air in buildings [5]. LPCBs can be metabolized in humans and diminished in the environment and are therefore not detectable with ambient monitoring. HPCBs have more than five chlorine atoms and are typically represent environmental exposure via the food chain [6]. The focus of this study according to LPCBs as well as HPCBs is the degree of chlorination. The chemical structure is focused in the third group,  the coplanar dlPCBs with a similar chemical structure to dioxins.”

Line 74 – the levels quantified in this cited study are not defined clinically as hypo or hyper thyroid.

Yes you are right. Bloom et al. (2014) only report correlations of different types of PCBs with different thyroid hormone related outcomes. We now phrased this part more precisely. See lines 80-84: “Various studies reported that PCB exposure alters thyroid function in both directions. Bloom et al. report a negative association of thyroid-like PCBs (28, 52, 60, 74, 77, 95, 99, 101, 105, 114, 118 and 126) with total T3 as well as fT4 [20]. Also lower levels of fT3 and fT4 in PCB were reported for exposed humans compared to a non-exposed control group [21]. In contrast positive associations were reported between different PCB congeners and fT4 in fish eaters [22].”

Line 77 – cite hypothyroid and depression study

Now we cited the following study. See lines 85-87. “The thyroid function is associated with depression [24]. Thus, a lower level of thyroid hormones is associated with more depressive symptoms.”

Figure 3 – axis of fT4 should be reversed so that low is on left.

Thank you. Yes you are right, the interpretation of the interaction-figures will be more intuitive, when the axis of fT4 will be reversed. However, we analyzed it with R and to the best of our knowledge, there was no possibility to change the direction of the axis. Maybe you have a sugestion. Please see following the R-command that we used for creating the figure.

##LPCB*fT4-->Depression

library(multilevel)

data(SDDlong)

library(lattice)

DxfT4_N=lm(sum_PHQD~Ln_Lip_NPCB+fT4+Ln_Lip_NPCB*fT4, data=SDDlong, control=list(opt="optim"))

summary(DxfT4_N)

TTM<-seq(min(SDDlong$Ln_Lip_NPCB),max(SDDlong$Ln_Lip_NPCB),length=25)

TTV<-seq(min(SDDlong$fT4, na.rm=TRUE),max(SDDlong$fT4, na.rm=TRUE),length=25)

TDAT2<-list(Ln_Lip_NPCB=TTM,fT4=TTV)

grid<-expand.grid(TDAT2)

fit<-predict(DxfT4_N,grid)

trellis.device(theme="col.whitebg")

wireframe(fit~Ln_Lip_NPCB*fT4,data=grid,col="steelblue4",

          screen=list(z=70,x=-55),

          xlab=list("LPCBs

                    (ln transformed)",cex=1.0),

          ylab=list("fT4 (pmol/L)",cex=1.0),ylim=c(12,22),

          zlab=list("Depressive

symptoms",cex=1.0),scales=list(arrows=F),

          zlim=c(-2,27), shade=T,colorkey=F)

Reviewer 2 Report

General Comments:

This manuscript evaluates the associations between plasma PCBs (differentiated into lower-chlorinated PCB congeners (LPCBs); higher chlorinated  PCB congeners (HPCBs); dioxin-like PCBs (dlPCBs); and hydroxy-PCBs, metabolites primarily of LPCBs) and urinary homovanillic acid, corrected for creatine content; and depressive symptoms in a group of occupationally exposed males (predominantly) at 3 different points in time. Strengths of the study are evaluation of real occupational exposures for associations of various classes of PCBs with depression; the statistical analyses of the data; the assay of different PCB congeners in the blood of the subjects; and the study population, many of who were analyzed at each time point.

Limitations include the speculated linearity of the interactions shown in Figure 1. The authors might have included a comparison of the relative binding affinities and plasma concentrations of the significantly most important congener (or congeners) of each class of PCB including HO-PCBs to transthyretin (TTR), in addition to the free thyroxin content (fT4) in the same plasma samples. Optimistically, this might help the authors understand why more of their data does not support their postulated mechanism than does.

Some other, often minor,  issues are described in more detail below.

Specific Comments:

L.20: --- it bounds binds to

L.24: --- Hhomovanillic acid

L.33: This statement suggests that psychological processes are not dependent upon physiological mechanisms. Evidence for this novel concept must be discussed with supporting references in the Introduction and Discussion of this manuscript. Otherwise this statement must be removed from the Abstract and anywhere else it occurs in the paper.

L.50: --- five and or more chlorine substituents ---

L.75: --- hypo- orand hyperthyroidism ---

L.87: --- albumin (5%) [25]).

L.91: --- Some PCB congeners have a structure similar to T4. Structure in this sense infers 3-dimensional structure and conformation. Part of the interest here should also be in the PCB congeners most closely linked with depression and their mechanism of action. If some of this information is known, it should be included in the Introduction. In this case which of the individual hydroxy-PCB congeners have highest affinity for binding to transthyretins.

L.110: Is it really a high PCB body burden or a high PCB blood content that is important here? PCBs in low turnover fat stores seems unlikely to be important in this context. The authors are encouraged to comment on this question, with references.

L.132: By the Ethics Commission of the Medical Faculty of each study center? If this is the case, should be stated? Is there a central university or faculty of medicine that served as the budget center for this research? If so, this center should be specifically identified as such along with the ethics review statement.

Table 1: Footnote 6 should be footnote 5.

L.164: What were the conditions of enzymatic hydrolysis: including enzyme and time. How was it determined that total/all glucuronide or sulphate conjugates of the OH-PCBs were hydrolyzed under these conditions?

L.165: As the methodology used for the analysis of the OH-PCBs is vey important to this study, some details are required. For example, what was the % recovery of free and conjugated OH-PCBs when they were run through this system in the same laboratory at the same time as this study was performed?

L.175: Some details of the electrochemiluminiscent immunoassay are required. It does not suffice to give a reference.

L.179: Unfortunately, urinary homovanillic acid is not a selective biomarker for brain dopamine metabolism. This must be addressed here. For example, dopamine is an important neurotransmitter in the kidney and in other areas of the peripheral nervous system.

Tale 2: How do you account for the fact that with t1: LPCB*fT4 but not t2 or t3, there is a significant association (P<0.001) with depressive symptoms but not with HVA/creatinine (P= .38)? Does this not suggest a limitation in the working hypothesis? Could peripheral HVA formation account for these differences?

L.340-343: Unfortunately, more of the results presented in this paper do not support “a physiological mechanism via blocked thyroid 340 transporter TTR and disturbed DA system to explain the positive correlation between PCB body 341 burden and depressive symptoms”. The authors should be very careful about cherry-picking their data to support their hypothesis.

Author Response

Dear Reviewer 2,

we thank you very much for your helpful comments regarding our submission. We have revised the manuscript based on the valuable suggestions and advice you made. An item-by-item response to your comments is enclosed. We hope that these revisions successfully address your concerns and requirements.

We thank all of you for your helpful suggestions and your time.

Sincerely,

Dr. Petra Gaum

Reviewer 2 – Item-by-Item-Response

Open Review

English language and style

( ) Extensive editing of English language and style required
( ) Moderate English changes required
(x) English language and style are fine/minor spell check required
( ) I don't feel qualified to judge about the English language and style

Yes

Can be   improved

Must be   improved

Not   applicable

Does the introduction provide sufficient background and include all   relevant references?

( )

(x)

( )

( )

Is the research design appropriate?

( )

(x)

( )

( )

Are the methods adequately described?

( )

( )

(x)

( )

Are the results clearly presented?

( )

(x)

( )

( )

Are the conclusions supported by the results?

( )

( )

(x)

( )

Comments and Suggestions for Authors

General Comments:

This manuscript evaluates the associations between plasma PCBs (differentiated into lower-chlorinated PCB congeners (LPCBs); higher chlorinated  PCB congeners (HPCBs); dioxin-like PCBs (dlPCBs); and hydroxy-PCBs, metabolites primarily of LPCBs) and urinary homovanillic acid, corrected for creatine content; and depressive symptoms in a group of occupationally exposed males (predominantly) at 3 different points in time. Strengths of the study are evaluation of real occupational exposures for associations of various classes of PCBs with depression; the statistical analyses of the data; the assay of different PCB congeners in the blood of the subjects; and the study population, many of who were analyzed at each time point.

Limitations include the speculated linearity of the interactions shown in Figure 1.

            We now changed the wording and phrase the figure description more careful.

See figure 1:

“Schematic illustration of the postulated interaction hypotheses.”

The authors might have included a comparison of the relative binding affinities and plasma concentrations of the significantly most important congener (or congeners) of each class of PCB including HO-PCBs to transthyretin (TTR), in addition to the free thyroxin content (fT4) in the same plasma samples. Optimistically, this might help the authors understand why more of their data does not support their postulated mechanism than does.

Unfortunately, TTR was not measured in the HELPcB program because the focus of the study was elsewhere. The tendency of PCBs to bind to TTR was also not measured in this study. Therefore, no conclusion can be drawn in this regard. This point is now be discussed in the discussion section. See lines 386-388: “Further, the affinity of PCB on TTR was not measured in the study population, so no it is not clear what PCB congeners have the highest binding affinity to TTR in our study population.”

Some other, often minor,  issues are described in more detail below.

Specific Comments:

L.20: --- it bounds binds to

            We now changed “bounds” into “binds”.

L.24: --- Hhomovanillic acid

            We now write homovanillic acid in small letters.

L.33: This statement suggests that psychological processes are not dependent upon physiological mechanisms. Evidence for this novel concept must be discussed with supporting references in the Introduction and Discussion of this manuscript. Otherwise this statement must be removed from the Abstract and anywhere else it occurs in the paper.

Thank you for pointing that out. Presumably we have not formulated this sentence precisely enough. The statement of this sentence should be that both psychological and physiological processes play a role in the development of depression after PCB exposure. We now excluded the information about psychological mechanisms, because they were not mentioned in the abstract or introduction before. This may be the reason for misunderstanding the mentioned sentence. We have now reformulated this sentence and hope that it will be easier to understand now. See lines 32-34 in the abstract: „The results give first hints that a physiological process involving the thyroid and DA system is responsible for depressive symptoms after PCB exposure.”

We have also changed this point in the conclusion section. See lines 404f: ”The findings show that there might be a physiological process involving the DA and thyroid system may be responsible for depressive symptoms after PCB exposure.”

L.50: --- five and or more chlorine substituents ---

            We now changed “and” into “or”.

L.75: --- hypo- orand hyperthyroidism ---

            We now changed “and” into “or”.

L.87: --- albumin (5%) [25]).

            The information of the reference is now given after the parenthesis.

L.91: --- Some PCB congeners have a structure similar to T4. Structure in this sense infers 3-dimensional structure and conformation. Part of the interest here should also be in the PCB congeners most closely linked with depression and their mechanism of action. If some of this information is known, it should be included in the Introduction. In this case which of the individual hydroxy-PCB congeners have highest affinity for binding to transthyretins.

Thyroid relevant PCB congeners are reported in the literature before. We now include information in the introduction as well as in the discussion section. See lines 80-82 in the introduction: „Bloom et al. report a negative association of thyroid-like PCBs (28, 52, 60, 74, 77, 95, 99, 101, 105, 114, 118 and 126) with total T3 as well as fT4 [20].”  And see lines 349-355 in the discussion: “Another possible reason that the interaction was only significant for LPCBs may be, that the group of LPCBs is the only PCB group in this study that only consists of thyroid-like PCBs [20]. Since the focus of this study was not solely on the thyroid gland, thyroid-like PCBs were not explicitly determined. Because of this no PCB with all thyroid-like congeners could be created.”

With regard to depression, no clear mechanism has yet been described in the literature and, accordingly, no specific congeners that are considered relevant to depression have yet been identified. One reason for this could be the multicausal predictors of depression.

L.110: Is it really a high PCB body burden or a high PCB blood content that is important here? PCBs in low turnover fat stores seems unlikely to be important in this context. The authors are encouraged to comment on this question, with references.

Thank you for pointing that out. The serum concentration of PCB correlates with the concentration in fatty tissue (r=.68; Mussalo-Rauhamaa, 1991). However, you are right about the considered mechanism. The PCB concentration in the blood is decisive for the postulated mechanism. We formulated this sentence more precisely now. See lines 119-121: „That means, a higher PCB and OH-PCB blood concentration should be accompanied with a negative association so that higher concentrations of fT4 are associated with lower HVA concentration.”

L.132: By the Ethics Commission of the Medical Faculty of each study center? If this is the case, should be stated? Is there a central university or faculty of medicine that served as the budget center for this research? If so, this center should be specifically identified as such along with the ethics review statement.

We now included the whole name and the missing information according to the study centers in the method section.

This study was approved by the Ethics Commission of the Medical Faculty of the RWTH Aachen University. All collaborating Institutes are part of the same faculty, except radiology in Dortmund. The radiology in Dortmund was a subcontractor und thereby even covered by the ethic vote. We now included this missing information. Information about funding was given in the document under “Funding” (see lines 412-414). The budget was provided by the Institution for Statutory Accident Insurance and Prevention in the Energy, Textile, Electrical, and Media Industry (BGETEM), Cologne, Germany. The budget was administered by the finance department of the University Hospital RWTH Aachen.

Table 1: Footnote 6 should be footnote 5.

            We now changed footnote 6 into footnote 5.

L.164: What were the conditions of enzymatic hydrolysis: including enzyme and time. How was it determined that total/all glucuronide or sulphate conjugates of the OH-PCBs were hydrolyzed under these conditions?

An aliquot of 100 µL of plasma together with 100 µL of ammonium acetate buffer 0.1    (pH= 5.3) and 5 µL of β- Glucuronidase/Arylsulfatase enzyme were used for the enzymatic hydrolysis, which was carried out overnight in a drying oven at 37 °C. The conditions have been shown enough for the release of the target compounds from plasma. Initial tests performed in the lab (data not published) with and without the use of the enzyme have shown that OH-PCBs levels in plasma were higher when the enzymatic hydrolysis was performed. 

L.165: As the methodology used for the analysis of the OH-PCBs is vey important to this study, some details are required. For example, what was the % recovery of free and conjugated OH-PCBs when they were run through this system in the same laboratory at the same time as this study was performed?

Recoveries for all compounds in plasma ranged from 71 to 134%, with an average exceeding 80%, which is available at Quinete et al.2015. Concentrations of total OH-PCBs in plasma have been assessed, which included not only the free phenolic forms but also the glucuronide and sulfate. Please note, this paper is not about method development, and more information on recoveries and method performance has already been published elsewhere.

We included a detailed description of OH-PCB detection as Appendix at the end of the manuscript. Please see Appendix B at page 13.

L.175: Some details of the electrochemiluminiscent immunoassay are required. It does not suffice to give a reference.

Yes you are right. We now included more information about the detection of fT4. First, a 15µl sample is incubated with a ruthenium complex of marked T4-specific antibodies. After adding biotinylated T4 and the reagent streptavidin-coated microparticles to form an antibody-hapten complex, the free binding sites of the marked antibody are occupied. The resulting complex is bound to the solid phase via the biotin-streptavidin interaction. The reaction mixture is then transferred to the measuring cell, where the microparticles are fixed to the surface of the electrode by magnetic action. Afterwards the unbound substances are removed with ProCell/ProCell M. The microparticles are then removed from the electrode. By applying a voltage, the chemiluminescence emission is induced and measured with the photomultiplier. The results are determined on the basis of a calibration curve.

L.179: Unfortunately, urinary homovanillic acid is not a selective biomarker for brain dopamine metabolism. This must be addressed here. For example, dopamine is an important neurotransmitter in the kidney and in other areas of the peripheral nervous system.

We discussed this point in the discussion section, but included this also in the method section now. See lines 215 - 216:“ Only 12% of urinary HVA originates from the brain [39], but nevertheless urinary HVA was used as indicator for DA, because of the non-invasive method and due to the examination design.”

Tale 2: How do you account for the fact that with t1: LPCB*fT4 but not t2 or t3, there is a significant association (P<0.001) with depressive symptoms but not with HVA/creatinine (P= .38)? Does this not suggest a limitation in the working hypothesis? Could peripheral HVA formation account for these differences?

One reason for this could be the multicausal etiology of depression (e.g. psychosocial factors), as well as the various influencing factors on urinary HVA (e.g. nutrition). In addition, the concentration of HVA shows a non-linear variation over the measurement occasions (Putschögl et al. 2015).

L.340-343: Unfortunately, more of the results presented in this paper do not support “a physiological mechanism via blocked thyroid 340 transporter TTR and disturbed DA system to explain the positive correlation between PCB body 341 burden and depressive symptoms”. The authors should be very careful about cherry-picking their data to support their hypothesis.

            We now rephrased this paragraph and used a more careful wording. See lines 396-399.

“The described results in this study are not clear enough to support a mechanism via blocked thyroid transporter TTR to explain depression after PCB exposure. But the results give first hints for a physiological mechanism via disturbed thyroid and DA system to explain the positive correlation between PCB body burden and depressive symptoms.”

Round 2

Reviewer 1 Report

I appreciate the author's thorough responses to my comments. They do a good job putting their results into more context to acknowledge potential limitations. These data are interesting and important to consider, given all the potential relationships between PCB effects on thyroid and DA function and depression.  I do, however, still have some concerns:

1. The positive association between fT4 and depression symptoms:
The single paper the authors cite linking thyroid function to depression (Hage and Azar, 2012) actually state that total and free T4 is often ELEVATED in patients with depression. (Another primary article I found shows a positive correlation between FT4 and depression severity - Berent et al 2014.) The rest of this review discusses other indicators of thyroid function (TSH, T3, rT3) and how these need to be considered in concert to get a true sense of thyroid function. For example, subclinical hypothyroidism is defined as elevated TSH and normal T3 and T4. Moreover, depression may be linked to normal peripheral thyroid function but isolated disruptions to central thyroid hormone action.  At the very least, the authors need to acknowledge that their single measure of thyroid function may not fully describe thyroid hormone functions in the brain.
Berent et al: https://www.ncbi.nlm.nih.gov/pubmed/244432282. The authors never explicitly state the assumption that, in people with low PCB exposure, serum fT4 is indicative of brain fT4 (which can then alter DA function and depression). If they made this statement explicit, and then contrasted it with the predicted association in people with significant PCB exposure, the source of the predicted interaction becomes more clear to me.

3. I agree that it is statistically appropriate to analyze and consider an interpretation without significant main effects. However, it seems to me that it is inappropriate to make the statement "When PCBs bind TTR, T4 cannot bind to it and will result in a higher level of free T4 in the blood" (Line 116-177) if this is not strongly supported in the current study (Table R1). I realize that it has been observed previously (Xu et al 2015), but the authors rely on this relationship to predict a positive association between fT4 and depressive symptoms in high-PCB exposed individuals but not others.  As the least, the authors should publish Table R1, so that readers can decide for themselves if these seems like a plausible explanation for the observed statistical interaction.

Author Response

Dear Reviewer 1,

again, we thank you very much for your helpful comments regarding our resubmission. We have revised the manuscript again. . An item-by-item response to your comments is enclosed. We hope that these revisions successfully address your concerns and requirements.

We thank you for your helpful suggestions and your time.

Sincerely,

Dr. Petra Gaum

Item-by-item response:

1. The positive association between fT4 and depression symptoms:
The single paper the authors cite linking thyroid function to depression (Hage and Azar, 2012) actually state that total and free T4 is often ELEVATED in patients with depression. (Another primary article I found shows a positive correlation between FT4 and depression severity - Berent et al 2014.) The rest of this review discusses other indicators of thyroid function (TSH, T3, rT3) and how these need to be considered in concert to get a true sense of thyroid function. For example, subclinical hypothyroidism is defined as elevated TSH and normal T3 and T4. Moreover, depression may be linked to normal peripheral thyroid function but isolated disruptions to central thyroid hormone action.  At the very least, the authors need to acknowledge that their single measure of thyroid function may not fully describe thyroid hormone functions in the brain.
Berent et al: https://www.ncbi.nlm.nih.gov/pubmed/24443228

Thank you very much for your comment. Now we have included the findings of Berent et al. (2014) in the introduction. See lines 91ff: “[…] Berent et al. [25] reported a positive association of fT4 with the improvement of depression. However, thyroid hormones had been elevated in studies with depressive humans [25,26].”

We also discussed the fact that other parameters than T4 must be considered for an explanatory mechanism by thyroid function. See lines 428-431 in the discussion section: „ Besides fT4 other thyroid related parameters might be relevant to get a clearer picture of the proposed pathomechanism [24]. It might be necessary to also investigate T3 or thyroid stimulating hormone, to describe a complete mechanism explaining depressive symptoms after PCB exposure. Future studies should take take these additional hormones into account in their analyses.“

2. The authors never explicitly state the assumption that, in people with low PCB exposure, serum fT4 is indicative of brain fT4 (which can then alter DA function and depression). If they made this statement explicit, and then contrasted it with the predicted association in people with significant PCB exposure, the source of the predicted interaction becomes more clear to me.

Thank you very much for that comment. We have now added the information for fT4 in the introduction and tried to illustrate the contrast between non-PCB-exposed and PCB-exposed persons. We hope that the postulated interaction is now easier to understand. Please see lines 88f: „ In non-PCB exposed humans, previous studies reported that serum fT4 can be an indicator for fT4 concentration in the brain [23].”

See also lines 120-129: “With regard to non PCB-exposed humans, and consideration of past literature, we assume that there is a positive association between ft4 and the main DA metabolite HVA. In case of high PCB blood concentration a negative association is postulated. Therefore an interaction hypothesis with opposite directions of the simple slopes were postulated. The negative association between fT4 and HVA exposure is expected in case of high PCB blood concentration means, that a higher PCB and OH-PCB blood concentration should be accompanied with a negative association; so that higher concentrations of fT4 are associated with lower HVA concentration. In contrast, a positive association between fT4 and HVA is expected in case of no PCB exposure. Humans with low or no PCB body burden should show a positive association that a low fT4 level is accompanied with lower HVA concentration.”

3. I agree that it is statistically appropriate to analyze and consider an interpretation without significant main effects. However, it seems to me that it is inappropriate to make the statement "When PCBs bind TTR, T4 cannot bind to it and will result in a higher level of free T4 in the blood" (Line 116-177) if this is not strongly supported in the current study (Table R1). I realize that it has been observed previously (Xu et al 2015), but the authors rely on this relationship to predict a positive association between fT4 and depressive symptoms in high-PCB exposed individuals but not others.  As the least, the authors should publish Table R1, so that readers can decide for themselves if these seems like a plausible explanation for the observed statistical interaction.

We have now removed the mentioned sentence. Further, we inserted table R1 into the method section. Please note that table R1 is now table 3 in the manuscript. See also lines 265-268: „According to Baron and Kenny [60] no direct effect between the predictor and the mediator is necessary to test an interaction hypothesis. Nevertheless, we provide the direct associations between several PCB congeners as well as OH-PCBs with fT4 in table 3.”

Reviewer 2 Report

The authors have carefully considered each of my original comments. In my opinion. this manuscript should be accepted for publication in the IJERPH.

Author Response

Dear Reviewer 2,

Thank you very much for this positive response related to the revised version of our manuscript.

We thank you for your helpful suggestions and your time.

Sincerely,

Dr. Petra Gaum